# Probabilistic Prediction Algorithm for Cycle Life of Energy Storage in Lithium Battery

**Xue Wang [1,*], Chunbin Gao [1] and Meng Sun [2]**

[1]   College of Electrical Engineering, Jilin Engineering Normal University, Changchun 130052, China; gaochunbin86@163.com

[2]   College of Food Engineering, Jilin Engineering Normal University, Changchun 130052, China; sunmengmeng88@163.com

*   Correspondence: wangxue8601@163.com; Tel.: +86-15948312719

**Abstract:** Lithium batteries are widely used in energy storage power systems such as hydraulic, thermal, wind and solar power stations, as well as power tools, military equipment, aerospace and other fields. The traditional fusion prediction algorithm for the cycle life of energy storage in lithium batteries combines the correlation vector machine, particle filter and autoregressive model to predict the cycle life of lithium batteries, which are subjected to many uncertainties in the prediction process and to inaccurate prediction results. In this paper, a probabilistic prediction algorithm for the cycle life of energy storage in lithium batteries is proposed. The LS-SVR prediction model was trained by a Bayesian three-layer reasoning. In the iterative prediction phase, the Monte Carlo method was used to express and manage the uncertainty and its transitivity in a multistep prediction and to predict the future trend of a lithium battery's health status. Based on the given failure threshold, the probability distribution of the residual life was obtained by counting the number of particles passing through the threshold. The wavelet neural network was used to study the sample data of lithium batteries, and the mapping relationship between the probability distribution of the residual life of lithium batteries and the unknown values were established. According to this mapping relation and the probability distribution of the residual life of lithium batteries, the health data could be deduced and then iterated into the input of the wavelet neural network. In this way, the predicted degradation curve and the cycle life of lithium batteries could be obtained. The experimental results show that the proposed algorithm has good adaptability and high prediction efficiency and accuracy, with the mean error of 0.17 and only 1.38 seconds by average required for prediction.

**Keywords:** lithium battery; cycle life; probability prediction; uncertainty; residual life; wavelet neural network

## 1. Introduction

With the worsening of the energy crisis and environmental problems in recent years, the new energy industry has developed rapidly, especially in the battery industry. Lithium batteries have been widely used in various fields, such as electronic products, electric tools, electric vehicles and energy storage fields because of their high energy density, no memory effect, small self-discharge and long cycle life. The research and development of power batteries with high energy density and long cycle life have become hot issues in the field of electric vehicle (EV). Based on the current research progresses and the accumulation of experience, lithium batteries with capacities above 300 Wh/kg can be realized [1,2]. However, at a specific point in time, the problem of energy storage will be inevitably encountered, i.e., the battery performance will continue to decline with the cycling and material aging until the battery is discarded [3]. In addition, the degradation of the battery performance cannot be

directly measured but needs to be estimated in advance to decide whether to replace the battery to avoid some unnecessary events. The performance of batteries can be divided into two categories: electrical performance and reliability. Battery life is one of the important indicators to measure the electrical performance of batteries. Charge–discharge cycles include a charge operation and a discharge operation. The number of charge–discharge cycles that a battery can carry out while maintaining a certain output capacity is called the cycle life (service life) of the battery. For energy storage batteries, it is generally believed that the life of the battery is terminated when the available capacity of the battery decreases to 70% of the initial level [4]. Battery life includes cycle life and calendar life, where the former refers to the number of cycles of the battery from a certain charging and discharging system to the end of life and the latter refers to the time required for the battery to be stored in a certain state until the end of life. There are many complex physical and chemical reactions in the charge and discharge processes of Li-ion batteries, and many factors can affect the cycle life of Li-ion batteries. On the other hand, cycle life testing is often time-consuming and costly. The correct evaluation of battery life is of guiding significance in the production and development of Li-ion batteries and affects the health management system of batteries.

The third industrial revolution proposed by Jeremy Rifkin is actually a new energy revolution, which significantly drives the development of smart grid. Among them, energy storage control technology is the core technology supporting the development of smart grid, and the system-on-chip technology is the core technology of energy storage control technology. State of Health (SOH) is a byte that must be attached to ensure the normal and flexible transmission of information payload for the network to operate, manage and maintain OAM (Operation, Administration and Maintenance). However, with the increasingly widespread use of lithium batteries, SOH estimation and life prediction technology will become an important core technology in the future. The prediction accuracy of lithium battery life will directly affect the maintenance plan and maintenance cost of users. Related literatures point out that battery life prediction methods can mainly be categorized into experience-based methods and performance-based methods [5,6]. Experience-based methods include the cycle number method, the ampere–time method, the weighted ampere–time method and the event-oriented aging accumulation method [7]. Performance-based methods include mechanism-based predictions, feature-based predictions and data-driven predictions [8]. The test of mechanism-based prediction is rather complicated as it is difficult to establish a perfect aging model. The measurement of the feature-based prediction is even more complex as it needs special measuring instruments. If the Electrochemical Impedance Spectroscopy (EIS) impedance spectrum is carried out online rapid measurement, further researches are needed. Relatively speaking, the testing method of data-driven predictions is more simple, which does not need to consider the mechanism and characteristics of the complex physical and chemical evolution process of lithium batteries. The fitting life curve of lithium batteries can be obtained by using the mathematical statistics method or artificial neural network model on the basis of test data. However, this method is also subjected to some limitations such as the requirement for large amounts of data accumulation and a long time for data accumulation. The artificial neural network model can well predict the life of lithium batteries in the early stage but is not suitable for predicting the life of lithium batteries in the late stage, which indicates the complexity of lithium batteries' aging process. Battery health is related to battery operation, performance, cycle life observation and so on. No one estimation method has the absolute advantage in precision for all states of lithium batteries [9]. The training dimension of the artificial neural network model is much more complex, and limited data will bring about considerable deviation. Due the differences in usage occasions, the cycle life of lithium batteries used in different energy storage systems will be different. Apparently, the reliability of accuracy and the accuracy are doubtful when using the same fitting parameter set, which brings great challenges to the development of lithium battery life prediction technology.

The Bayesian LS-SVR method derives model parameters by introducing a multilayer Bayesian probabilistic reasoning framework based on the basic LS-SVR algorithm. While shortening the modeling

time, the framework makes the LS-SVR model have the ability to generate probabilistic predictive output. In order to improve the accuracy and practicability of prediction, it is necessary to combine the internal structure of the lithium battery with the external working environment so that a suitable method can be obtained for predicting the cycle life of the energy storage in a lithium battery [10]. This will be conducive to expressing and managing the inherent uncertainty in the prediction issue.

In 1992, Zhang Qinghua and Benvenist Albert proposed the wavelet neural network (WNN) by wavelet expansion and translation, which is a feedforward neural network with the wavelet element function as the neuron function. The wavelet neural network has a good approximation ability, adaptive ability and information fusion ability of nonlinear systems [11]. In recent years, the wavelet neural network has been extensively used in many fields, such as fault detection and identification, traffic prediction and intelligent optimal control.

To solve the shortcomings of traditional prediction algorithms for the cycle life of lithium batteries, a probabilistic cycle life prediction algorithm for lithium batteries was proposed. The algorithm draws lessons from the successful application of the LS-SVR method in the time-series prediction and uses Bayesian framework to express the uncertainty of the model. The Monte Carlo method was used to transfer the uncertainty and to calculate the probability distribution of the residual life. Combining the idea of a cyclic multistep prediction with the single-step prediction of the wavelet neural network, the mapping relationship between the probability distribution of the residual life of lithium batteries and the unknown values was established. By comparing the predicted value with the failure criterion, the life end point is determined and a more accurate probability prediction of battery cycle life is realized.

This paper introduces two prediction methods, namely the probability prediction algorithm of lithium battery residual life based on the Bayesian LS-SVR and the prediction algorithm of lithium battery cycle life based on the wavelet neural network. To analyze the prediction performance of the newly proposed probabilistic prediction algorithm of lithium battery cycle life, a comparative study was conducted among the proposed algorithm, the prediction algorithm based on the Mean Impact Value (MIV) Back Propagation (BP) neural network and the algorithm based on the correlation vector machine particle filter and autoregressive model. Finally, the application of the proposed algorithm in lithium battery life prediction as well as its practicability, efficiency and precision were analyzed.

## 2. Materials and Methods

*2.1. Probabilistic Prediction Algorithm for the Cycle Life of Energy Storage in a Lithium Battery*

2.1.1. Probabilistic Prediction Algorithm for the Residual Life of a Lithium Battery Based on the Bayesian LS-SVR

The prediction framework for the residual life of a lithium battery based on the Bayesian LS-SVR is shown in Figure 1.

A capacitor is used to discharge the battery at a constant current. The capacity of the battery is equal to the time of discharge times the discharge current. Like most methods, the SOH is characterized using the capacity in this paper:

$$SOH = \frac{C_i}{C_o} \times 100\% \tag{1}$$

where $C_o$ is rated capacity and $C_i$ is the capacity of the *i*th charge–discharge cycle. When the capacity is reduced to 70% of the rated capacity, the lithium battery function is judged to be invalid.

(1) Training sample selection

When training samples are few, all historical data can be used as training samples [12,13]. When the sample size is large, the rolling time window method is used to extract the training data in view of the demand of the online prediction. Assuming that the length of the window is n, the latest n data at any current time is selected as training data. This method has a small amount of computation, which

is a significant advantage compared with others. Moreover, the model training in this method can also be focused on learning the latest evolution trend of lithium battery health.

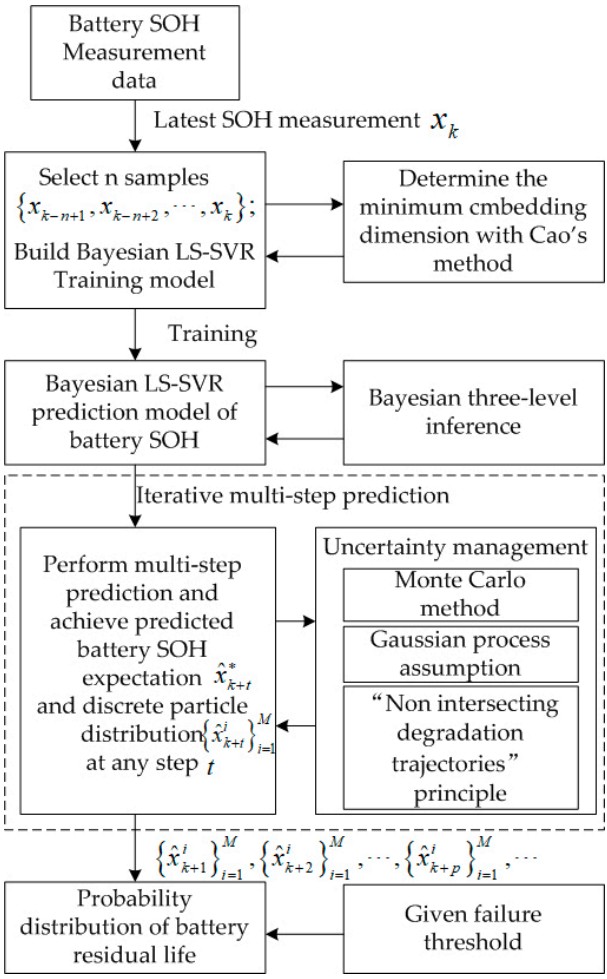

**Figure 1.** The prediction framework for the residual life of a lithium battery based on the Bayesian LS-SVR. SOH is State of Health.

(2) Phase space reconstruction

The selected training sample sequence of lithium battery's health status is $\{x_{k-n+1}, x_{k-n+2}, \cdots, x_k\}$. Takens' Theorem is a basic proposition for calculating the dimension of the embedded phase space, which is also the theoretical basis of phase space reconstruction technology. An important problem in chaotic applications is to reconstruct an n-dimensional phase space containing the chaotic motion from the time series of a single variable. According to Takens' Theorem [14], the inherent evolution rule can be restored in a high dimensional space, that is, when the embedding dimension $d$ of the phase space is greater than a certain value, there exists a smooth mapping $f$, making the time-series delay time $\tau$ Equation (1). Then, there is

$$x_{k+1} = f([x_k \quad x_{k-1} \quad \cdots \quad x_{k-d+1}]) \tag{2}$$

Equation (2) is a one-step prediction formula in which the high-dimensional mapping function $f$ can be trained by the Bayesian LS-SVR method. In order to reduce the computational complexity, it is necessary to determine the minimum embedding dimension $m$. In this paper, Cao's method was used to determine such a minimum embedding dimension $m$. [15]. The $i$th phase point in the phase space can be expressed as $x_i(d) = [x_i \quad x_{i+1} \quad \cdots \quad x_{i+d-1}]$, where the nearest neighbor point is represented by $x_{(i,d)}(d)$, as shown by

$$a(i,d) = \frac{\|x_i(d+1) - x_{(i,d)}(d+1)\|}{\|x_i(d) - x_{(i,d)}(d)\|} \tag{3}$$

where $i = 1, 2, \ldots, n - d$.

$$E(d) = \frac{1}{n-d} \sum_{i=1}^{N-d} a(i,d) \tag{4}$$

where $E(d)$ is the function of the embedding number $d$, which can be expressed as follows:

$$E_1(d) = \frac{E(d+1)}{E(d)} \tag{5}$$

When the value is greater than $d_0$, $E_1(d)$ no longer changes significantly but approaches to 1, then $m = d_0 + 1$ is the minimum embedding dimension that is looked for.

(3) Bayesian LS-SVR Model Training

After determining the minimum embedding dimension, the training of the model can be started.

(i)　Data preprocessing: In order to obtain more accurate prediction results, all input and output data are normalized before being used for training, i.e., converted to the value range of [0,1]. According to the degradation characteristics of a lithium battery's health state,

$$g(x) = x^* = \frac{x - x_{\min}}{x_{\max} - x_{\min}} \tag{6}$$

where $x_{\max}$ and $x_{\min}$ are the maximum and minimum values of the training samples respectively.

(ii)　To establish training sample pairs: At the current $k$ moment, $n$–$m$ training samples pairs can be obtained according to the selected samples and the minimum embedding dimension.

$$\begin{cases} X = \begin{bmatrix} x_1 \\ x_2 \\ \vdots \\ x_{n-m} \end{bmatrix} = \begin{bmatrix} x_{k-n+1} & x_{k-n+2} & \cdots & x_{k-n+m} \\ x_{k-n+2} & x_{k-n+3} & \cdots & x_{k-n+m+1} \\ \vdots & \vdots & \cdots & \vdots \\ x_{k-m} & x_{k-m+1} & \cdots & x_{k-1} \end{bmatrix} \\ y = \begin{bmatrix} y_1 \\ y_2 \\ \vdots \\ y_{n-m} \end{bmatrix} = \begin{bmatrix} x_{k-n+m+1} \\ x_{k-n+m+2} \\ \vdots \\ x_k \end{bmatrix} \end{cases} \tag{7}$$

(iii)　Bayesian LS-SVR model training: According to the training steps of the Bayesian three-layer reasoning, the kernel parameter $\sigma_0$ and regularization parameter $\gamma_0$ are initialized first, then a linear search is performed from $\sigma_0$ until finding $\sigma_{MP}$, which makes the $p(D \mid \sigma)$ be the max in the third layer reasoning. Then $\gamma_{MP}$ and the LS-SVR model are derived.

$$y_{MP} = \omega_{MP}^T \varphi(x_{test}) + b_{MP} \tag{8}$$

(4) Iterative Prediction and Uncertainty Management

The life of Li-ion batteries is defined to be terminated when the actual discharge capacity of the Li-ion batteries is lower than 70%.

Generally, there are two kinds of prediction methods for the Li-ion battery life cycle [16]: direct predictions and iterative predictions. Since the time range of lithium batteries entering the failure state is uncertain and unknown, direct prediction is not feasible. The Bayesian LS-SVR prediction model obtained in the previous section is used to iteratively predict the SOH of lithium batteries until

it crosses the failure threshold. The Bayesian LS-SVR prediction model has the ability to obtain the probability distribution output. However, the residual life prediction is a multistep prediction problem, which needs to consider the transfer of the prediction uncertainty between steps [17], that is, how the uncertainty of the previous prediction output is accurately transmitted as input to the next step and ultimately to the predicted output. In this paper, the Monte Carlo method was used to realize the representation and management of the prediction uncertainty in the multistep prediction [18]. The Monte Carlo method is a random simulation method, which is also known as a statistical simulation or random sampling technology. It is a calculation method based on probability and statistical theory, which uses random numbers or more common pseudorandom numbers to solve many computational problems. As shown in Figure 2, the starting time of the prediction is assumed to be $k$.

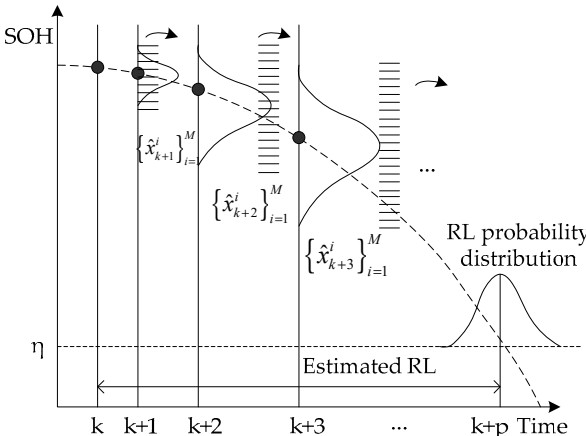

**Figure 2.** The multistep iterative prediction and Monte Carlo sampling based on the Bayesian LS-SVR.

In the first step, $x_{n-m+1} = \begin{bmatrix} x_{k-m+1} & x_{k-m+2} & \cdots & x_k \end{bmatrix}$ is the input and $\hat{x}_{k+1}$ and $\hat{\sigma}_{k+1}$ are obtained from the expected value and difference of the probabilistic predictive output, where

$$\hat{x}_{k+1} = \omega_{MP}^T \varphi(x_{n-m+1}) + b_{MP} \tag{9}$$

In the second step, the Gauss distribution $N\left(\hat{x}_{k+1}, (\hat{\sigma}_{k+1})^2\right)$ is sampled by the Monte Carlo method. All particles are sorted from large to small, so as to get the set $\left\{\hat{x}_{k+1}^i\right\}_{i=1}^M$ where $M$ is the number of sample particles. Then, $M$ input vectors $\left\{x_{n-m+2}^i = \begin{bmatrix} x_{k-m+2} & \cdots & x_k & \hat{x}_{k+1}^i \end{bmatrix}\right\}_{i=1}^M$ are constructed, the Bayesian LS-SVR prediction model is input and the outputs of M particles and M Gaussian density distribution $N\left(\hat{x}_{k+2}^i, \left(\hat{\sigma}_{k+2}^i\right)^2\right)$ are obtained.

$$\hat{x}_{k+2}^i = \omega_{MP}^T \varphi\left(x_{n-m+2}^i\right) + b_{MP} \quad (I = 1, 2, \cdots, M) \tag{10}$$

It can be seen that the output of the second step prediction is equal the weight Gauss mixture distribution, that is,

$$q(\hat{x}_{k+2}) = \frac{\sum\limits_{i=1}^M N\left(\hat{x}_{k+2}^i, \left(\hat{\sigma}_{k+2}^i\right)^2\right)}{M} \tag{11}$$

From the second step, it can be seen that the prediction uncertainty in the multistep prediction consists of two parts: One is the uncertainty of the prediction model itself, corresponding to the Gaussian output of each particle, and the other is the uncertainty introduced when using the previous step prediction output as the next step prediction input, which is manifested in the mixture of Gaussian distribution.

In the third step, since $q(\hat{x}_{k+2})$ is a Gaussian mixture distribution composed of M Gaussian distributions, it is difficult to obtain a satisfactory discrete approximation if the distribution is sampled with M as the sample number. If the number of samples is increased, the number of samples will increase exponentially in the later iteration, which is not feasible in long-term predictions. In this paper, the deterioration of the lithium battery's health status is considered as a Gaussian process. A normal distribution $N\left(\hat{x}_{k+2}^{*}, \left(\hat{\sigma}_{k+2}^{*}\right)^{2}\right)$ of equal mean and equal force-difference is used to approximate $q(\hat{x}_{k+2})$, where

$$\hat{x}_{k+2}^{*} = E(\hat{x}_{k+2}) = \frac{\sum\limits_{i=1}^{M} \hat{x}_{k+2}^{i}}{M} \tag{12}$$

$$\left(\hat{\sigma}_{k+2}^{*}\right)^{2} = E\left(\left(\hat{x}_{k+2}\right)^{2}\right) - \left(E(\hat{x}_{k+2})^{2}\right) = \\ \frac{\sum\limits_{i=1}^{M}\left(\left(\hat{x}_{k+2}^{i}\right)^{2}+\left(\hat{\sigma}_{k+2}^{i}\right)^{2}\right)}{M} - \frac{\left(\sum\limits_{i=1}^{M} \hat{x}_{k+2}^{i}\right)^{2}}{M^{2}} \tag{13}$$

Then the Gauss distribution $N\left(\hat{x}_{k+2}^{*}, \left(\hat{\sigma}_{k+2}^{*}\right)^{2}\right)$ is sampled by the Monte Carlo method, and all the $M$ particles are sorted from large to small to obtain set $\left\{\hat{x}_{k+2}^{i}\right\}_{i=1}^{M}$. Then the $M$ input vectors $\left\{\hat{x}_{n-m+3}^{i} = \left[x_{k-m+3} \cdots x_{k} \hat{x}_{k+1}^{i} \hat{x}_{k+2}^{i}\right]\right\}_{i=1}^{M}$ are established, and the next Gaussian mixture distribution is obtained by the input prediction model.

In step $p$ ($p > 3$), the step 3 for the iterative prediction is repeated until all particles pass through the failure threshold or complete the specified prediction step.

In order to establish the particle sequence connection between steps, it is necessary to sort the particles according to an identical rule after each sampling. In this paper, it is assumed that the principle of "degenerate trajectories do not intersect each other" is satisfied [19], i.e., the relationship between the degradation magnitude of each branch track remains unchanged at any time. As shown in Figure 3, if the relationship between any two particles at an $k + 1$ time satisfies $\hat{x}_{k+1}^{i} \geq \hat{x}_{k+1}^{j}(1 \leq i, j \leq M, i \neq j)$, their subsequent sub-particles at any time also satisfy $\hat{x}_{k+t}^{i} \geq \hat{x}_{k+t}^{j}(t > 1)$. This principle reflects the trend consistency of each branch's degeneration trajectory derived from the same degeneration trajectory in the future, which is consistent with the actual situation and is beneficial to reducing the uncertainty of the prediction.

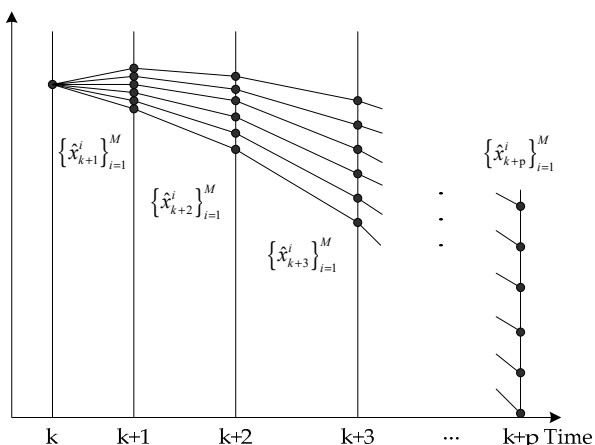

**Figure 3.** A schematic diagram of the disjoint principle of the degenerate trajectories.

(5) Probability Distribution Approximate of Residual Life

According to Section 2.1.1 (4), the approximate distribution $\left\{\hat{x}_{t}^{i}\right\}_{i=1}^{M}$ of the health state of lithium batteries can be obtained at any time $t$ within a given prediction step. Firstly, the inverse function

$g^{-1}$ of Equation (6) is used to inversely transform $\left\{\hat{x}_t^i\right\}_{i=1}^M$ and obtain the actual health state value $\left\{\tilde{x}_t^i\right\}_{i=1}^M$. Combined with the failure threshold η, there are two methods for calculating the residual life of lithium batteries. (I) The bilateral failure definition [20] is used to count the number of particles in the symmetrical region above and below the failure threshold, so that the probability distribution of the residual life can be calculated. The weight of each particle is $1/M$. (II) Since the health degradation process of lithium batteries is a monotonic process in theory, the cumulative failure probability at time $t$ can be expressed as follows according to the single threshold failure definition [21]:

$$F(t) = \mathrm{Pr}(\tilde{x}_t \leq \eta) = \int_0^\eta q(\tilde{x}_t)dx_t \tag{14}$$

where $\mathrm{Pr}(\bullet)$ is the probability of satisfying the condition.

For the Monte Carlo approximation, there is

$$F(t) \approx \frac{\tilde{x}_t^i number : \tilde{x}_t^i \leq \eta}{M} \quad (i = 1, 2, \cdots, M) \tag{15}$$

Therefore, the probability distribution of residual life is

$$r(t) = F(t) - F(t-1) \tag{16}$$

### 2.1.2. Prediction of the Lithium Battery Cycle Life Based on the Wavelet Neural Network

Cycle life refers to the number of charge–discharge cycles that the battery can withstand before its capacity decreases (attenuates) to a certain specified value under a certain charge/discharge current profile. In this paper, the charge/discharge current profile is set as 10 C. Before using the wavelet neural network to predict, the parameters of the network need to be set [22]. According to the demand of the cyclic multistep prediction, the number of output nodes is selected to 1, and then the wavelet basis function as well as the number of input layer and hidden layer nodes are discussed. Appropriate settings can be obtained according to the probabilistic distribution of the residual life of lithium batteries based on the Bayesian LS-SVR. The number of nodes in the input layer and hidden layer is set to be within a suitable range. The effects of different wavelet basis functions on the sample processing of lithium batteries are compared [23]. Using the same method, the number of input layer nodes and hidden layer nodes can be determined [24].

The number of input layer nodes was set to 6 and that of hidden layer nodes was set to 8. The convergences of the training error in the WNN training with respect to the Mexican Hat wavelet, the Marr wavelet and the Morlet wavelet as wavelet basis functions were compared. It can be seen that the convergence speeds of the 3 wavelet basis functions are similar with each other. In order to obtain the best prediction effect, the wavelet basis function with the least convergence error should be selected, that is, the Morlet Wavelet should be selected. Similarly, the number of input layer nodes was set to 4 and that of the hidden layer nodes was set to 6.

The wavelet neural network has a strong stability. In structural design, it avoids the appearance of the local optimal problem and has a strong function learning ability. Therefore, the wavelet neural network is used to predict the cycle life of batteries. After completing the WNN network setup, it can complete the network learning by using sample training. The life cycle data of battery No. 1 was selected as the training samples to train the neural network, and then the cycle life of batteries No. 2–4 were predicted. Generally, the failure of a battery is defined when its capacity is deteriorated by 70% [25–27]. However, in this paper, the failure of battery is reached when battery capacity is deteriorated by 25%; that is, when SOH = 75%, the failure of battery occurs.

The cycle life of No. 2 battery was predicted when the SOH of the batteries decreased to 90%, 85% and 80%. The predicted results are shown in Table 1.

**Table 1.** The probability prediction results of the No.2 lithium battery cycle life.

| Forecast Starting Point | SOH = 90% | SOH = 85% | SOH = 80% |
|---|---|---|---|
| predicted value | 150 | 155 | 159 |
| actual cycle life | 159 | 160 | 172 |
| relative error/% | 5.7% | 3.1% | 7.6% |

Among them, the relative error formula is as follows:

$$\varphi = \frac{|L_r - L_p|}{L_r} \times 100\% \tag{17}$$

where $L_r$ refers to the actual cycle life of the battery and $L_p$ refers to the battery's predicted cycle life.

## 3. Results

The probabilistic prediction algorithm presented in this paper is used to predict the cycle life of energy storage in lithium batteries. The results are shown in Tables 2–4.

**Table 2.** Probability prediction results of No. 3 lithium battery cycle life.

| Forecast Starting Point | SOH = 90% | SOH = 85% | SOH = 80% |
|---|---|---|---|
| predicted value | 150 | 151 | 154 |
| actual cycle life | 159 | 159 | 159 |
| relative error/% | 5.7% | 5.1% | 3.1% |

**Table 3.** Probability prediction results of No. 4 lithium battery cycle life.

| Forecast Starting Point | SOH = 90% | SOH = 85% | SOH = 80% |
|---|---|---|---|
| predicted value | 109 | 154 | 152 |
| actual cycle life | 165 | 162 | 168 |
| relative error/% | 33.9% | 4.9% | 9.5% |

**Table 4.** Probability prediction results of No. 18 lithium battery cycle life.

| Forecast Starting Point | SOH = 90% | SOH = 85% | SOH = 80% |
|---|---|---|---|
| predicted value | 130 | 163 | 152 |
| actual cycle life | 180 | 180 | 168 |
| relative error/% | 27.8% | 9.4% | 9.5% |

It can be seen from Tables 2–4 that the predicted value is more accurate when the suitable starting point is chosen in the midlife of lithium batteries. The starting point of prediction when SOH = 85% is more suitable. The predicted error of No. 4 lithium battery at SOH = 85% is larger than that of No. 3 lithium battery and No. 18 lithium battery at SOH = 85%, but the change is not significant. It is worth noting that the predicted error of the No. 4 lithium battery is larger than that of the No. 3 lithium battery and No. 18 lithium battery at SOH = 90%. This is due to the fact that the SOH value of the No. 4 lithium battery produces a drastic change around 90%, especially the early change has a great impact on the prediction. Moreover, a more accurate prediction of the cycle life can be obtained in the later prediction.

The algorithm cannot effectively predict the change of SOH but has a good adaptability to the change of the actual value. Tests were conducted during four different periods, as shown in Figure 4 and Table 4. During the mid-cycle life, the measured value of the capacity of the No. 4 lithium battery changed sharply during the 40th cycle and 46th cycle, resulting in a sharp rise (i.e., a sharp change) of SOH value. In the 65th cycle and the 75th cycle, the measurements of the capacity of the No. 4 lithium

battery showed wavelet changes, and the SOH value increased slightly. The prediction error of the cycle life of the No. 4 lithium battery during this period was large, with the prediction starting point being 39 times, 40 times and 46 times. Fitting linear attenuation curves before and after the change point are shown in Figure 4 and Table 5. The health level represents the surplus usage of the battery.

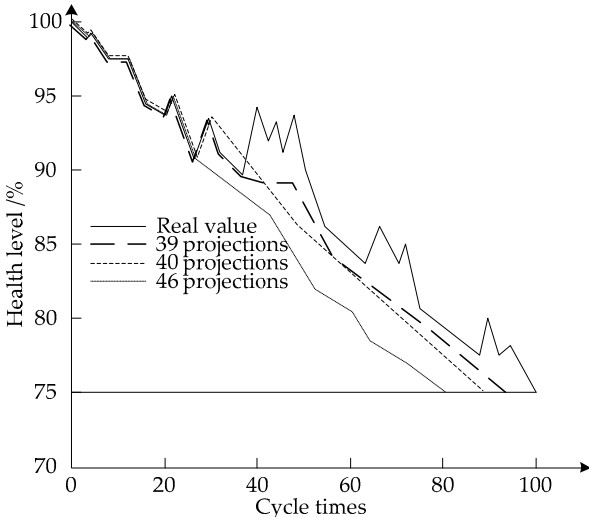

**Figure 4.** Fitting linear attenuation curves before and after the change point.

**Table 5.** Comparison of the prediction results before and after the battery change point.

| Forecast Starting Point | The Thirty-Ninth Cycle | The Fortieth Cycle | The Forty-Sixth Cycle |
|---|---|---|---|
| predicted value | 150 | 150 | 168 |
| actual cycle life | 180 | 176 | 175 |
| relative error/% | 16.67% | 14.77% | 4.00% |

From Figure 4 and Table 5, it can be seen that the proposed algorithm has a good adaptability to change. The prediction values before and after change were quite different. The prediction from the change point was significantly more accurate than that before the change. If the capacity of lithium batteries is observed to have a change, the prediction should be conducted again in order to obtain a reliable cycle life prediction value.

In this paper, the experiment was conducted based on "battery data set 1" published by NASA Arms Research Center. These data were collected by a self-defined battery prediction test bench, which consists of a commercial lithium battery typed by 1850, a programmable DC electronic load, a power supply, a voltmeter, an ammeter, a thermocouple sensor, an electrochemical impedance spectrometer and a PXI cabinet for data acquisition and experimental control. Four lithium batteries (No. 5, No. 6, No. 7 and No. 18) were repeatedly operated in charging mode, discharging mode and impedance measurement mode at room temperature, respectively. The operation of charging was carried out under a constant current of 1.5 A until the lithium battery voltage reached 4.2 V and stopped when the charging current dropped to 20 mA. The operation of discharging was carried out at a constant current of 2 A until the voltage of the lithium battery dropped to 2.7 V, 2.5 V, 2.2 V and 2.5 V, respectively; the impedance measurement was performed by frequency scanning within the range of 0.1–5000 Hz with an electrochemical impedance spectroscopy detector. The cyclic charge–discharge operation led to the accelerated degradation of the lithium battery performance. When the capacity of the lithium battery was reduced to 70% of the rated capacity (from 2.0 to 1.4 Ahr), the battery was defined to be invalid and the test was stopped. The capacity degradation data of two energy storage lithium batteries, No. 5 and No. 7, are shown in Figure 5.

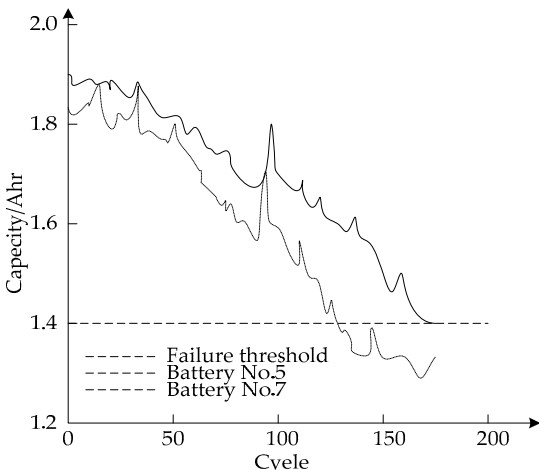

**Figure 5.** The capacity degradation trajectories of No.5 lithium battery and No.7 lithium battery.

As can be seen from Figure 5, the capacity of lithium batteries did not present a perfect monotonic decreasing trend due to self-charging, especially near the 90th cycle where a large fluctuation was observed. The degradation trajectories of the two lithium batteries did not conform to the bi-exponential model in the previous literature [28]. Moreover, the degradation trajectories and cycle life of the two lithium batteries were still quite different under similar experimental conditions, and the battery capacity did not decrease with the change of environment. Therefore, it was difficult to establish other unified and accurate degradation models. The experimental results show that the proposed algorithm has high practicability in predicting the cycle life of lithium batteries.

The efficiency of the proposed algorithm was experimentally verified by measuring the budget lifecycle time of energy storage in lithium batteries. The annual working power consumption of the battery was calculated, and the estimated service life of the battery was calculated according to the residual capacity of 10%. By using fifteen lithium-ion batteries as experimental objects, the cycle life of lithium batteries was predicted by using the proposed algorithm, BP neural network life prediction algorithm for LiFePO4 battery based on MIV, correlation vector machine, particle filter and autoregressive model fused algorithm. The comparison of the prediction time of different algorithms is shown in Table 6.

**Table 6.** The comparison of the cycle life budget (s) of energy storage in a lithium battery by using different algorithms.

| Lithium Battery Number | Algorithm in This Paper (s) | Life Prediction Algorithm of LiFePO4 Battery Based on BP Neural Network Based on MIV (s) | Prediction Algorithm for Residual Life of Li-ion Batteries Based on Correlation Vector Machine Particle Filtering and Self-Regression Model (s) |
|---|---|---|---|
| 1 | 1.35 | 1.78 | 1.73 |
| 2 | 1.38 | 1.74 | 1.84 |
| 3 | 1.38 | 1.78 | 1.81 |
| 4 | 1.41 | 1.82 | 1.85 |
| 5 | 1.43 | 1.83 | 1.85 |
| 6 | 1.37 | 1.95 | 1.91 |
| 7 | 1.38 | 1.81 | 1.84 |
| 8 | 1.43 | 1.86 | 1.88 |
| 9 | 1.36 | 1.87 | 1.91 |
| 10 | 1.41 | 1.84 | 1.87 |
| 11 | 1.38 | 1.83 | 1.87 |
| 12 | 1.37 | 1.85 | 1.87 |
| 13 | 1.35 | 1.77 | 1.82 |
| 14 | 1.35 | 1.86 | 1.88 |
| 15 | 1.35 | 1.86 | 1.88 |
| Average value | 1.38 | 1.83 | 1.85 |

Three algorithms in Table 6 were used to estimate the cycle life of lithium batteries. The results are shown in Figure 6.

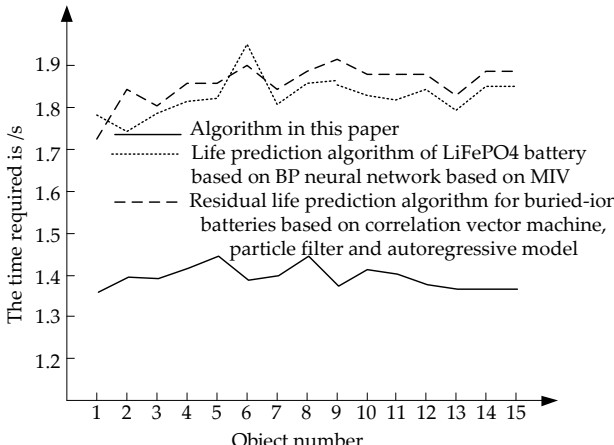

**Figure 6.** The comparison of time required for the prediction of energy storage in a lithium battery cycle life t when using three different algorithms.

As shown in Table 6, the time required to predict the cycle life of lithium batteries using the proposed algorithm was shorter, and all of them were completed within 1.41 s. The life prediction algorithm of a LiFePO4 battery based on the MIV BP neural network was completed within 1.82 s. The prediction algorithm for the residual life of a lithium ion battery based on the correlation vector machine particle filtering and self-regression model was completed in 1.85 s. With the increase of the number of experimental subjects, the time required for predicting the cycle life of lithium batteries tended to be stable, which was maintained at 1.35 s, with the average value of 1.38 s. The time required for the algorithm based on MIV to predict the cycle life of a lithium iron phosphate battery fluctuated from 1.74 to 1.95 s, with an average value of 1.83 s. In contrast, the prediction algorithm based on the correlation vector machine particle filtering and self-regression model took the longest time for prediction, which was 1.85 seconds by average. Through a comprehensive analysis based on Tables 5 and 6, it can be known that the proposed algorithm takes the least time and achieves the highest efficiency in predicting the cycle life of lithium batteries.

Aging generally occurs as the battery has been recharged and cycled multiple times after the battery has been filled with liquid. Normally, it can be divided into normal temperature aging or high temperature aging. The aging effect is to stabilize the properties and composition of the solid electrolyte interface (SEI) films formed after the first charge. The aging temperature at room temperature is 25 °C, and sometimes it can be 38 or 45 °C.

Aging is usually predicted to accelerate between 70% and 80% SOH. In this study, the prediction accuracy of the LiFePO4 battery life prediction algorithm based on the MIV BP neural network was compared with that of the Lithium ion battery life prediction algorithm based on the correlation vector machine particle filter and autoregressive model when the cycle life of a lithium battery occurs between 70% and 80% SOH, as shown in Figure 7.

Figure 7 shows that when SOH is 70, the accuracy of the proposed algorithm, the prediction algorithm based on the MIV BP neural network and the algorithm based on the correlation vector machine particle filter and autoregressive model are 82%, 42% and 30%, respectively. When SOH is 76%, the accuracy of the proposed algorithm, the prediction algorithm based on the MIV BP neural network and the algorithm based on the correlation vector machine particle filter and autoregressive model are 84%, 43% and 20%, respectively. When SOH is 80%, the accuracy of this algorithm, the prediction algorithm based on the MIV BP neural network and the algorithm based on the correlation vector machine particle filter and autoregressive model are 82%, 42% and 18% respectively. It can be

concluded that the accuracy of the proposed algorithm has a higher prediction accuracy and stable prediction performance compared with the other two algorithms.

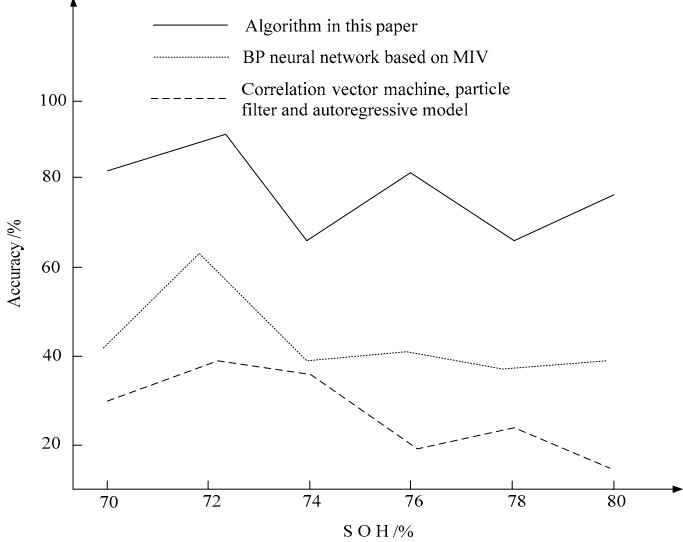

**Figure 7.** The comparison of the life prediction accuracy of three algorithms between 70% and 80% SOH.

In order to test the performance of the algorithm, 10 groups of lithium batteries with different cycle lives were predicted under the same experimental environment, as shown in Tables 6–8. The predicted results of the different algorithms were compared with the actual cycle life of lithium batteries, as shown in Tables 7–9.

**Table 7.** The prediction results of the proposed algorithm.

| Object Number | Actual Measurement Cycle Life (Times) | The Measuring Life of the Algorithm in This Paper (Times) | Error Value (Times) |
|---|---|---|---|
| 1 | 8.5 | 8.3 | 0.2 |
| 2 | 13.5 | 13.3 | 0.2 |
| 3 | 20.4 | 20.3 | 0.1 |
| 4 | 31.3 | 31.5 | 0.2 |
| 5 | 40.3 | 40.4 | 0.1 |
| 6 | 49.2 | 49.6 | 0.4 |
| 7 | 58.5 | 58.5 | 0 |
| 8 | 65.5 | 65.5 | 0 |
| 9 | 72.6 | 72.9 | 0.3 |
| 10 | 80.1 | 80.3 | 0.2 |
| Average value | | | 0.17 |

**Table 8.** The prediction results of the LiFePO4 battery life prediction algorithm based on the MIV BP neural network.

| Object Number | Actual Measurement Cycle Life (Times) | Life Prediction Algorithm of LiFePO4 Battery Based on MIV BP Neural Network (Times) | Error Value (Times) |
|---|---|---|---|
| 1 | 8.1 | 8.6 | 0.5 |
| 2 | 13.5 | 13.7 | 0.2 |
| 3 | 20.2 | 20.1 | 0.1 |
| 4 | 31.3 | 31.9 | 0.8 |
| 5 | 40.4 | 40.0 | 0.4 |
| 6 | 49.2 | 49.6 | 0.4 |
| 7 | 58.7 | 58.2 | 0.5 |
| 8 | 65.5 | 65.3 | 0.2 |
| 9 | 72.1 | 72.8 | 0.7 |
| 10 | 80.1 | 80.9 | 0.8 |
| Average value | | | 0.46 |

**Table 9.** A detection system for the residual life prediction of lithium-ion batteries based on the correlation vector machine, particle filter and autoregressive model.

| Object Number | Actual Measurement Cycle Life (Times) | Residual Life Prediction Algorithm of Lithium Ion Battery Based on Correlation Vector Machine Particle filter and Autoregressive Model (Times) | Error Value (Times) |
|---|---|---|---|
| 1 | 8.1 | 8.9 | 0.8 |
| 2 | 13.2 | 13.9 | 0.7 |
| 3 | 20.4 | 20.9 | 0.5 |
| 4 | 31.0 | 31.3 | 0.3 |
| 5 | 40.1 | 40.6 | 0.5 |
| 6 | 49.2 | 49.7 | 0.5 |
| 7 | 58.3 | 58.9 | 0.6 |
| 8 | 65.4 | 65.9 | 0.5 |
| 9 | 72.1 | 72.9 | 0.8 |
| 10 | 80.1 | 80.6 | 0.5 |
| Average value | | | 0.57 |

In order to compare the average error values of the three algorithms more clearly, the average error values obtained by the different algorithms are represented by histograms, as shown in Figure 8.

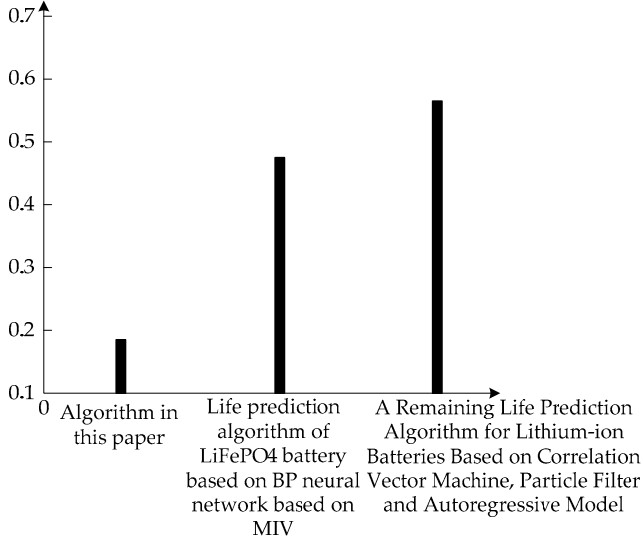

**Figure 8.** The average error values of the different algorithms.

As shown in Tables 7–9, the average error value of the proposed algorithm, the prediction algorithm based on the MIV BP neural network and the algorithm based on the correlation vector machine particle filter and autoregressive model are 0.17%, 0.46% and 0.57%, respectively. The error value of the proposed algorithm is significantly smaller than that of the other two algorithms. The experimental results show that the proposed algorithm has a higher accuracy in predicting the cycle life of lithium battery.

## 4. Discussion

In this paper, the Bayesian LS-SVR algorithm is applied to the prediction of the cycle life of lithium batteries. The performance of the proposed algorithm was investigated from different aspects through experimental analyses.

### 4.1. Analysis of the Discontinuous Changes in the Life Prediction of Energy Storage in a Lithium Battery

From the experiment on the discontinuous changes in the probability prediction of the cycle life of lithium batteries, it can be seen that the predicted value is more accurate when the suitable starting point is chosen in the midlife of the lithium battery, the relative error of the cycle life prediction is less than 10%,

and the proposed algorithm has a good adaptability to the discontinuous changes. This is due to the fact that the proposed algorithm is a probabilistic prediction algorithm, which can effectively manage the uncertainties generated in long-term predictions and its transmission in multistep predictions and thus can obtain the probability distribution of the cycle life of energy storage in lithium batteries.

### 4.2. Analysis of the Practicability of the Proposed Algorithm

From the experiment on the practicability of the probabilistic algorithm in predicting the cycle life of lithium batteries, it can be seen that the degradation trajectory and cycle life of the two lithium batteries are quite different. Compared with the previous prediction algorithm, the proposed algorithm is more practical. The reason is that this algorithm is a pure data-driven algorithm, which does not need a large number of historical data and model offline training. The proposed algorithm is flexible, versatile and suitable for the cycle life prediction of lithium batteries.

### 4.3. The Efficiency and Accuracy of the Proposed Algorithm

From the experiment on the accuracy of the probabilistic prediction algorithm in predicting the cycle life of lithium batteries, it can be seen that compared with the prediction algorithm based on the MIV BP neural network and the algorithm based on the correlation vector machine particle filter and autoregressive model, the proposed algorithm has a higher prediction efficiency and accuracy. This is mainly due to the fact that the proposed algorithm combines the Bayesian LS-SVR algorithm (predicting the probability distribution of the residual life of lithium batteries through model training, iterative operation and uncertainty management) and the wavelet neural network (determining the number of input nodes and hidden nodes of the training model), which saves lots of prediction time.

## 5. Conclusions

In this paper, a probabilistic prediction algorithm for the cycle life of lithium batteries is proposed through combining the probabilistic prediction algorithm based on the Bayesian LS-SVR with the prediction algorithm based on the wavelet neural network. The advantage of the proposed algorithm is that the neural network can be trained by using the life-cycle data of one battery without the need to update the model parameters online after training. At the same time, the prediction algorithm proposed in this paper has a good adaptability to the discontinuous changes occurring in the actual value. As long as the appropriate prediction starting point is selected, more accurate prediction results of the cycle life can be obtained. It is worth noting that the proposed algorithm took a very short time to finish the prediction, which is normally completed within 1.41 s. With the increase of the number of experimental objects, the time required for a prediction by the proposed algorithm tends to be stable, and the average error is only about one third of that of the traditional algorithm. The experimental data shows that the proposed algorithm can predict the cycle life of a lithium battery more accurately and more efficiently, which lays a foundation for the further research in the battery industry.

**Author Contributions:** Conceptualization, X.W. and C.G.; methodology, X.W.; software, M.S.; validation, X.W., C.G. and M.S.; formal analysis, X.W.; investigation, M.S.; resources, C.G.; data curation, X.W.; writing—original draft preparation, X.W., C.G. and M.S.; writing—review and editing, X.W.; visualization, M.S.; supervision, C.G. and M.S.; project administration, X.W.; funding acquisition, X.W.

**Funding:** This research received no external funding.

**Conflicts of Interest:** The authors declare no conflict of interest.

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
