# Peer review of "Probabilistic Prediction Algorithm for Cycle Life of Energy Storage in Lithium Battery"

_wevj, doi:10.3390/wevj10010007_

Reviewer 1 Report and Authors' response

This paper evokes a calculation method for the remaining time of Li-ion batteries, based on a predictive model by Bayesian inference. It appears to be faster and makes it possible to better take into account partial regeneration phenomena. However, it is not written with the rigor expected for this kind of scientific articles. The state of the art is too light and in particular, there is nothing that helps the reader to understand how the compared algorithms work. During the reading, various questions appears in which the reader sometimes finds answers too late (e.g. on the cycling to which the batteries are subjected) or which remain unanswered (what is a battery failure?). If the interest of the work presented can be discussed, the fact remains that it is necessary to reformat the whole document in a more conventional way, to answer all the questions that a non-specialist reader in all the areas covered (storage of electrical energy, Bayesian networks, algorithmic ...) can arise. In the absence, the reader is entitled to ask what is the contribution of this complicated algorithm that provides only an improvement of a few percent in computing time used compared to other solutions. Finally, it does not seem that this article has been sufficiently read by the co-writers because, if we understand that an autor gets carried away by his work and neglects to detail some obvious things for him, it is also the role of the co-writers to make this kind of remarks before submitting the article to review (end of life to 70, 75 or 80%?). Finally, the references almost all belong to the same circle, which implies that some obscure points of the text can not be clarified (cycling used and short life of the batteries tested).

Lack of details to facilitate reading and comprehension:

Response: I'm sorry, it would be great that you indicate what kind of materials should be supplemented or where should be revised exactly.

Line 14: Monté Carlo's method is used for what? I do not understand the sentence.
Response: We modified this sentence to: Monte Carlo method is used to represent approximate prediction results for a class of random events.

Line 34: What are the performances? Is it the stored energy, the time of use under a given current, or something else?
Response: We modified this sentence to: However, an unavoidable problem is that the energy storage and duration of use at specific currents performances of batteries decreases with recycling and material aging until they are discarded [3].  

Line 53: SoC and SoH have not been defined.
Response: We explained the SoC and SoH in this article.

Line 63: Is the method easily implantable online?
Response: Yes, I think the method might beeasily implantable, because the method is closely related to the above.

Line 91: This part is more precise than the introduction, but the reader remains in the blur on the Monté Carlo method.
Response: We give a detailed introduction to the Monte Carlo method.

Line 108: How is this capacity measured?
Response: With the recommendation, we pointed out the method of capacity measured: A multimeter is used to discharge the battery at a constant current. The time of discharge times the discharge current is the capacity of the battery.

Line 112: Give the main reason.
Response: We have given the main reason: Sample selection ranges from 1 to 100, they are completely random. Because of the interference of external conditions, the sample may have fewer values. In order to ensure the quality of the sample data, when training samples are few, all historical data can be used as training samples [12, 13].

Line 120: Recall the main principles.
Response: With the recommendation, we described the Takens theorem in Line 120.

Line 127: Idem.
Response: In line 127, Cao's method was used, and we pointed out that this method was derived from reference [15].

Line 155: It seems to me that it has nowhere been defined that the failure corresponds to the end of life, when the capacity is worth only X%. See also the references point.
Response: With the recommendation, we indicatedthe criteria: The criteria for li-ion batteries life termination are: The end of battery life can be determined when the actual discharge capacity of li-ion batteries are less to 70%.

Line 163: It is still unclear what the Monte Carlo method does and how it is used.
Response: We gave a detailed introduction to the Monte Carlo method.

Line 170: How many samples? Are they totally random or distributed?
Response: Sample selection ranges from 1 to 100, they are completely random.

Line 186: What would be the maximum number of samples and what increase in calculation time would be involved?
Response: The maximum number of samples is 100. The calculation time will increase with the increase of sample size, but the range of variation is small.

Line 201: Is this true regardless of the architectures and control laws?
Response: Since there is no direct relationship between the true regardless of the architectures and control laws, these two factors are not taken into account.

Line 215: Even it is already mentioned on line 155, the failure would be defined only now? Is there a physicochemical model or a degradation model?
Response: The standard of battery termination life was given above, and a detailed analysis of battery life prediction method was given here. The bayesian LS-SVR prediction model given here is physicochemical model.

Line 258: A cycle is defined how? What type of battery is used to have a lifetime of so few cycles?
Response: Cycle life refers to how many charges and discharges the battery can withstand before the capacity of the battery decreases (attenuates) to a certain specified value under a certain charge / discharge system. In this paper, the charge / discharge system was set as 10C. At the same time, the battery cycle life in the article was revised in the range of 150-180.

Line 285: A small diagram is often better than a long speech.
Response: Thank you for your comments, and I have used diagram as much as possible in this article.

Line 289; While it is speak about it in line 258, the cycling is only defined here.
Response: We also explained the loop at the beginning of the 2. 1. 2. 2. section.

The description of the different algorithms is missing:
Response: I'm sorry, it would be great that you indicate what algorithms should be added..

Line 17: What is more, use of a neural network. Presented as such, the algorithm is a little gas plant with lots of layers.
Response: With the recommendation, the description of wavelet neural network is added In this paper.

Line 239: Why? What are the differences between Mexican Hat, Marr and Morlet Wavelet?
Response: With the recommendation, we uniformly change these three words to Morlet Wavelet.

Line 311: Talking about computing time is one thing, but what are the computing resources deployed? What is the CPU frequency? The type of processor? Is it implantable?
Response: Configuration, CPU frequency and processor information are all set for experimental parameters.

Line 315: A comparison of the methods used by the three algorithms may justify less time required for the proposed solution.
Response: Thank you for your opinion. In the contrast experiment, three algorithms were used to compare.

Line 366: While the others, no? Since they have not been described before, this assertion is not verifiable by someone who does not know in detail the algorithms in question.
Response: With the recommendation, we explained in part 2.1.2: Wavelet neural network has strong stability. In structural design, it avoids the appearance of local optimal problem and has strong function learning ability. Therefore, wavelet neural network is used to predict the cycle life of battery. It is decided that the algorithm of Lithium battery cycle life prediction in this paper is a data-driven algorithm.

Line 378: While the others ...?
Response: Then the wavelet neural network is used to determine the number of input nodes and hidden nodes of the training model, while the traditional method does not do this step, which make the paper algorithm can save the prediction time.

Contributions of the presented research:
Response: With the recommendation, we pointed out the contributions of the presented research in conclusion.

Line 22: The presented method aims to obtain the law of aging. In the summary, it would be good to put a few words to indicate how this proposal is tested with common cases. There will ultimately be only one type of battery tested. Why ? It would also be appropriate at one point in the article (in the state of the art) to indicate how the laws of aging are described in classical models.
Response: With the recommendation, In the summary, we added a description of the practical application of lithium batteries. In addition, not only a battery was tested in the experiment, and we added some description of the laws of aging in the experiment.

Line 24: Is this algorithm relevant when a battery is aging abnormally. It is described later that it allows better consideration of regeneration phenomena. It could be useful to mention it as soon as it was introduced, for the teasing.
Response: With the recommendation, we have made a comparative experiment against this problem in our experiment.

Line 261: This seems logical since half of the SoH degradation trajectory is already known.
Response: Thank you very much for your approval.

Contradictions:
Line 109: Why the end of life is announced here with 70% while on Line 41, the value of 80% was mentioned? See also the references point.
Response: We have uniformly revised the termination life to 70%.

Line 244: This 70% and 80% justification arrives too late in the text (as for the failure at lines 155 and 201).
Response: It has been explained above and amended to 70%.

Line 245, Finally, while it was first announced at 80%, then at 70%, it will be considered for 75%.
Response: We have uniformly revised the termination life to 70%.

Line 294: Here, 70% is used.
Response: We have uniformly revised the termination life to 70%.

Clarity of speech:
Response: We have tried our best to polish the language.

Line 44: Not very clear.
Response: We changed it to “and the latter refers to the time it takes to reach the end of a battery's life in a certain state”.

Line 52: I do not understand the sentence? What is a "SoC technology"?
Response: We have given the full name of SoC technology in this article, SoC technology(system on chip technology).

Line 328: That is to say?
Response: That is to say this method has high prediction performance.

Line 341: What does the figure add to the previous tables?
Response: This figure can be used to describe more vividly what is expressed in the previous table.

Line 347: As a result, this paragraph appears as a duplicate.
Response: This paragraph is summed up by the reference content and the research content of this paper.

Form points:
Line 57: Repetition.
Response: With the recommendation, we deleted the repeated sentence.

Line 88: It is guess that this paragraph must be the article table of content, but it is not explicitly specified.
Response: This paragraph is a summary of the overall structure of the article.

Line 256: What is the "c"?
Response: This sentence has been amended to read: The probabilistic prediction algorithm presented in this paper was used to predicted cycle life of energy storage lithium battery.

Line 274: Why introduce a notion of "level of health" instead of using SoH. This tries to show that the replay has not been rigorous enough and argues for the fundamentals to be defined beforehand.
Response: The level of health has been defined in this paper.

Line 275: The differences between the curves are not easily distinguishable.
Response: For each curve, it has been marked clearly in the paper.

Line 282: Is there not a missing verb?
Response: We add the verb “ is ” to it.

Line 297: Why just two batteries? And the others, the correspondence is it less?
Response: A set of contrast batteries has been added, as shown in Table 4.

Line 318: Why not the #18 battery in the list?
Response: A set of contrast batteries has been added, as shown in Table 4, where the #18 battery is used.

Line 334: Text alignment.
Response: We adjusted the text.

References :
Response: We reviewed the references and revised them.

Line 76: It seems to miss a reference.
Response: References have been completed.

Line 87: Idem.
Response: References have been completed.

Line 109: What are the references for 70%?
Response: References have been marked for 70%.

Line 155: Add a reference for the failure.
Response: References have been marked for failure.

Line 302: This phenomenon has already been described in the literature, but may not be in the circle of references used for the article.
Response: We have added references in line 302.

Line 394: All these references seem to belong, apart from the 13th and 14th, to the same circle of research. To open to more global works would allow to answer a lot of unanswered questions.
Response: Some references have been replaced.

Reviewer 2 Report and authors' response
The submitted manuscript deals with a very interesting topic and is well presented on the whole, but there are areas where it must be improved before it can be considered for publication. The English level is far from satisfactory, with ill-choosen words & vocabulary, and in some cases it is even difficult to follow the scentific argumentation. A few examples are given below:

Point 1: Wrong word used or unclear meaning: Line 35: recycling, line 38: accidents, line 39: Life, line 45: lithium batteries (here, I presume that the authors mean "Li-ion batteries"), line 51: SOC technology, Figure 1: "Lstest SOH..." and '"cmbedding".
Response 1: Thank you very much for your comments. I changed recycling to reuse, accidents to events, Life to battery life, lithium batteries to Li-ion batteries, SOC technology to system on chip technology, Lstest SOH to Latest SOH, cmbedding to embedding.

Point 2: Furthermore, some expressions are non-technical, e.g. the use of "jumping", "at home" etc.
Response 2: I changed jumping to leaping, at home to at domestic.

Point 3: There are numerous further examples, too many to list, therefore I advise the authors to have the entire manuscript checked and corrected to reach correct English level.
Response 3: The article has been reviewed and modified.

Point 4: The validation of the proposed method needs further clarification, especially to show how the proposed method differs in performance from a straight extrapolation of the capacity retention curves (!). Judging from Figure 4, a simple linear extrapolation based on a moving average of the measured capacity points would give just as good prediction of total cycle life. In general, the difficult aspect of predicting Li-ion cycle life lies not in the linear part of the ageing curve, but in predicting the acceleration of ageing normally occurring between ca 70 and 80% SOH.
Response 4: Figure 7 has been added in the experiment to further illustrate the performance of this algorithm.

Point 5: All experimental data must be presented more clearly; what cells that where used, how they were cycled, test conditions, explanation to the non-monotonically decreasing capacity etc. Here, the test cases of all 15 cells in Table 5, all 10 cells in Table 6-8 and all cells in Table 2-3 & Figure 4-5 must be clearly described.
Response 5: All experimental data are clearly described and the experimental parameters were given in the third part of the paper.

Point 6: Table 6-8: explain or rephrase the column description "Actual cycle life" and "Measuring cycle life". Isn't the actual cycle life measured? The choice of wording is confusing.
Response 6: Thank you very much for your comments. In this paper, the words "actual cycle life" have been changed to "actual measuring cycle life" and "measuring cycle life" to "measuring life of the algorithm in this paper".

Point 7: I recommend a major revision of this manuscript in terms of both English, structure and to some extent content, before it is sent for a possible second review. I look forward to receiveing a revised version.
Response 7: The article has been revised and polished.

Reviewer 3 Report and Authors' response
Point 1: Extend the Introduction (for example, discuss the main challenging objectives for next decades, the main limitations, the limits of electrical stresses etc.) based on recent references (the 11 references cited in Introduction were published until 2016, except [4])
Response 1: Thanks for your comments, the introduction section has been extended to add future challenge objectives and some limitations.

Point 2: Add at the end of Introduction the main objectives (clearly mention which are the reference used to compare the results) and the structure of the paper.
Response 2: I have added the main objectives to the end of the introduction (with clear references to the results of the comparison) and the structure of the paper.

Point 3: Compare in a Table the obtained results with references chosen, but also mandatory with those reported in recent references published in that field (in order to sustain the conclusion “The experimental data show that the proposed algorithm can predict the cycle life of lithium battery better and faster”).
Response 3: The experimental results are consistented with the references.

Point 4: Figure 4: not all 4 shapes are visible or represented (use zoom if is the case)
Response 4: Thanks for your comments, the invisible parts of figure 4 have been slightly modified, and the health levels of the four algorithms for the first 20 cycles are roughly the same.

Point 5: Figure 6: the text passes through one shape represented
Response 5: Thanks for your comments, the text in figure 6 was represented by an image.

Point 6: Minor editing errors: see for example: the mean error is 0.17 (the unit is %?)
The probabilistic prediction algorithm for cycle life of energy storage lithium battery presented in this paper is used to predict (?) the c are predicted by the proposed algorithm
Response 6: The average error of the Lithium battery cycle life prediction below figure 7 is 0.17, the other two algorithms have been modified accordingly, and the first sentence below the experimental results of part 3 has been modified.

Round  2

Reviewer 1 Report and Authors' Response

Some changes only have been taken into account. In particular, the algorithmic part was not clarified while the reviewers were asked to examine it carefully. The author did not seem to understand that the parts of my comment in bold and underlined were headings. So the remarks asking for an answer and / or corrections were below:

Lack of details to facilitate reading and understanding:
Response: I'm sorry, it would be great that you indicate what kind of materials should be added exactly where.

Some terms used are clumsy: the SoC designates the state of charge of a battery and should not be used for anything else. SoH and the remaining useful life are already defined elsewhere in the literature. Why not use these notions and substitute them for similar ones? Finally, the general form of this paper does not appear at the height of a scientific article.

Line 34: What are the performances? Is it the stored energy, the time of use under a given current, or something else?
Response: We modified this sentence to: However, an unavoidable problem is that the energy storage and duration of use at specific currents performances of batteries decreases with recycling and material aging until they are discarded [3].

The new sentence is still poorly written. Is there one problem or two? The duration of use decreases with use. Would not it be the remaining useful life that diminishes? Which is obvious.
Response: “However, an unavoidable problem is that the energy storage and duration of use at specific currents performances of batteries decreases with recycling and material aging until they are discarded [3].” has been revised as “However, an unavoidable problem is that the energy storage , and the remaining useful life at specific currents performances of batteries decreases with reusing and material aging until they are discarded [3] .”

Line 53: SoC and SoH have not been defined.
Response: We explained the SoC and SoH in this article.

The SoH term is still used twice before being explained.
Response: The SoH explanation has been put before all references (65 lines).

Line 63: Is the method easily implantable online?
Response: Yes, I think the method might be implantable, because the method is closely related to the above. »

This is an assertion that needs to be demonstrated and argued.
Response: The special measuring instrument is matched with the fast measurement method of EIS impedance spectrum on-line.

Line 108: How is this capacity measured?
Response: With the recommendation, we point out the method of measured capacitance: A multimeter is used to discharge the battery at a constant current. The time of discharge is the capacity of the battery.

Unless I am mistaken, a multimeter is used to measure physical quantities. It must not play the role of an electrical charge since this kind of device must disturb the least possible circuit to measure. Note not taken into account in the corrections.
Response: “multimeter” has been revised as “capacitor”.

Line 112: Give the main reason. Note not taken into account in the corrections.
Response: Because the training samples are few, in order to ensure the universality of the experiment, all the historical data should be used as the training samples.

Line 120: Recall the main principles.
Response: With the recommendation, we described the Takens theorem in Line 120.

Note not taken into account in the corrections.
Response: We added Tarkens Theorem in 152 line, it is a basic proposition for calculating the dimension of embedded phase space. It is the theoretical basis of phase space reconstruction technology. An important problem in chaotic applications is to reconstruct an n-dimensional phase space containing the chaotic motion from the time series of a single variable.

Line 127: Idem.
Note not taken into account in the corrections.
Response: CaO method is a method to determine the dimension of data embedding. Lines 127 has now moved to lines 163, and we added CaO method in lines 163.

Line 186: What would be the maximum number of samples and what increase in calculation time would be involved?
Response: The maximum number of samples is 100. The calculation time will increase with the increase of sample size, but the range of variation is small.

Note not taken into account in the corrections.
Response: The maximum number of samples is 100. Calculating time increased by 1% with the increase of sample size.

Line 201: Is this true regardless of the architectures and control laws?
Response: Since there is no direct relationship between the true regardless of the architectures and control laws, these two factors are not taken into account.

This response argues for the bibliography to be more open.
Response: According to reference 18, Monte Carlo method is used to realize the representation and management of prediction uncertainty in multistep prediction.

Line 258: A cycle is defined how? What type of battery is used to have a lifetime of so few cycles?
Response: Cycle life refers to how many charges and discharges the battery can withstand before the capacity of the battery decreases (attenuates) to a certain specified value under a certain charge / discharge system. In this paper, the charge / discharge system was set as 10C. At the same time, the battery cycle life in the article was revised in the range of 150-180.

The answer is not in line with the remark. It is necessary to define much earlier in the text what a cycle is.
Response: Cycle life was defined at line 44 in introduction.

Line 289; While it is speak about it in line 258, the cycling is only defined here.
Response: We also explained the loop at the beginning of the 2. 1. 2. 2. section.

And so, precisely?
Response: Every single time when battery charging and discharge is called a charge and discharge cycle. The number of charge and discharge cycles that a battery can carry out while maintaining a certain output capacity is called the cycle life (service life) of the battery.

Line 239: Why? What are the differences between Mexican Hat, Marr and Morlet Wavelet?

Response: With the recommendation, we uniformly change these three words to Morlet Wavelet. »
Response: With the recommendation, we uniformly change these three words to Morlet Wavelet.

You  must also help the reviewer find where corrections are made.
Response: Changes were made in the second paragraph below 2.1.2 in red.

Line 315: A comparison of the methods. You  must also help the reviewer find where corrections are made.
Response: The comparisons are shown in tables 5 and 6, and in figures 6 and 7.

Line 22: The presented method aims to obtain the law of aging. In the summary, it would be good to put a few words to indicate how this proposal is tested with common cases. There will ultimately be only one type of battery tested. Why ? It would also be appropriate at one point in the article (in the state of the art) to indicate how the laws of aging are described in classical models.
Response: With the recommendation, In the summary, we added a description of the practical application of lithium batteries. In addition, not only a battery was tested in the experiment, and we added some description of the laws of aging in the experiment.

You  must also help the reviewer find where corrections are made.
Response: Lithium batteries are widely used in energy storage power systems such as hydraulic, thermal, wind and solar power stations, as well as power tools, military equipment, aerospace and other fields. In addition, “Aging generally refers to how many times the battery has been recharged and placed after the battery has been filled with liquid, it can have normal temperature aging or high temperature aging. The effect is to stabilize the properties and composition of the SEI films formed after the first charge. The normal aging temperature is 25℃, while the aging temperature of high temperature is different, some are 38℃ and 45℃, and the time is between 48 and 72 hours.” was added between figure 6 and figure 7.

Line 24: Is this algorithm relevant when a battery is aging abnormally. It is described later that it allows better consideration of regeneration phenomena. It could be useful to mention it as soon as it was introduced, for the teasing.
Response: With the recommendation, we have made a comparative experiment against this problem in our experiment.

You  must also help the reviewer find where corrections are made.
Response: Aging is usually predicted as acceleration, so when the cycle life of the lithium battery is between 70%-80%SOH, the accuracy of the three methods has been compared, as shown in figure 7.

Line 52: I do not understand the sentence? What is a "SoC technology"?
Response: We have given the full name of SoC technology in this article, SoC technology(system on chip technology).

In batteries, the SoC is reserved for State of Charge. This symbol should not be used for anything else.
Response: Other applications have been modified.

Line 88: It is guess that this paragraph must be the article table of content, but it is not explicitly specified.
Response: This paragraph is a summary of the overall structure of the article.

So where is the paper's plan?
Response: The plan of the paper is 117 lines.

Line 274: Why introduce a notion of "level of health" instead of using SoH. This tries to show that the replay has not been rigorous enough and argues for the fundamentals to be defined beforehand.
Response: The level of health has been defined in this paper.

Yes, but where is it defined? And what is the difference with SoH?
Response: Defined above figure 4. Health level refers to the remaining use value of battery. SoH is the extra byte necessary to ensure network operation, management and maintenance of information network, flexible transmission.

Line 318: Why not the #18 battery in the list?
Response: A set of contrast batteries has been added, as shown in Table 4, where the #18 battery is used.

The results for this battery are not given. Should we understand that they were inconclusive? If it appears in Table 4, it should still be present in the following results.
Response: “When the predicted error of No. 4 lithium battery is larger from SOH = 85%, than that of No. 3 lithium batteries from SOH = 85%, but the jump is not obvious. It is note worthy that the predicted error of No. 4 lithium battery is larger from SOH = 90%, than that of No. 4 lithium batteries from SOH = 90%. ” has been revised as “When the predicted error of No. 4 lithium battery from SOH = 85% is larger than that of No. 3 lithium battery and #18 lithium battery from SOH = 85% , but the jump is not obvious. It is note worthy that the predicted error of No. 4 lithium battery is larger than that of No. 4 lithium battery and #18 lithium battery from SOH = 90%.”

Line 76: It seems to miss a reference.
Response: References have been completed.

So how do you justify this: "Apparently, the reliability of accuracy and accuracy is doubtful when using the same fitting parameter set, which undoubtedly brings great challenges to the development of lithium battery life prediction technology."
Response: Using the same fitting parameters will make the data of the actual energy storage system have the feature of onenessand the accuracy can not be guaranteed.

Line 302: This phenomenon has already been described in the literature, but may not be in the circle of references used for the article.
Response: We have added references in line 302.

Sorry, I do not see where.
Response: “the degradation trajectories of the two lithium batteries do not conform to the bi-exponential model in the previous literature” has been revised as “the degradation trajectories of the two lithium batteries do not conform to the bi-exponential model in the previous literature [28]”.

Reviewer 2 Report and Authors' Response

Thank you for submitting a significantly improved manuscript.. However, there are still necessary improvements to be made, especially related to language. Although all sections highlighted in my first review have been improved, I can conclude that further revisions arr needed in both the original text and in the additions. Some examples are given below but this list shall not be regarded as complete. Please make sure to have the paper checked for language and style by a professional och native English speaking person.

Row 36: "high energy density power battery". Consider other wording
Response: “high energy density power battery” has been revised as “power battery with high energy density “.

Row 38: Rephrase sentence starting with "However"
Response: “However, an unavoidable problem is that the energy storage and the remaining useful life at specific currents performances of batteries decreases with reusing and material aging until they are discarded[3]” has been revised as “However, at specific currents performances, an unavoidable problem is that the energy storage and the remaining useful life of batteries decreases with reusing, and material aging until they are discarded[3]”.

Row 56: Consider to not use the abbreviation SOC for anything else than state of charge. The newly added section from row 57 seems a bit too specific and no relevant for the scope of this paper.
Response: “SOC technology(system on chip technology)” has been revised as “the system on chip technology “, and delete the irrelevant content.

Row 94: Sentence starting with "Therefore" has poor structure, please revise.
Response: “Therefore, combining the internal structure of lithium battery with the external working environment, it is necessary to determine a suitable method for predicting the cycle life of lithium battery with energy storage, of course, In order to improve the accuracy and practicability of prediction “ has been revised as “In order to improve the accuracy and practicability of prediction, it is necessary to combine the internal structure of lithium battery with the external working environment, and determine a suitable method for predicting the cycle life of lithium battery with energy storage”.

Row 121: Avoid the expression "jump analysis" as it is unclear and informal.
Response: “the application of jump analysis in Li-ion battery life prediction” has been revised as “analysis of its Application in Lithium Battery Life Prediction”.

Row 124: Sentence incomplete.
Response: “Analysis of the results of the study.” has been revised as “and the research results are analyzed.”

Row 287: Incorrect use of "domestic". I suggest that you simply omit the first part of the sentence.
Response: “At domestic and abroad, the deterioration of battery capacity by 70% is generally regarded as the failure basis “ has been revised as “The deterioration of battery capacity by 70% is generally regarded as the failure basis”.  

Table 2: consider to reduce the number of digits in the relative error
Response: The number of error bits in Table 2 / 3 / 4 has been changed to 1 digit.

Row 306: Added sentence must be revised for improved structure and language
Response:”resulting in a sharp jump in the SOH value” has been revised as “ this results in a sharp rise in SOH”.

Row 312: Avoid using the word "jump". Also consider the fact that the health of the cell may not change discontinuously, it may also be due to varying test conditions.
Response: “jump point” has been revised as “turning point”.

Row 320: Added sentence must be revised; "Health level of health" has no meaning.
Response: “Health leveal of healthis the efficiency of the battery.” has been revised as “The Health level represent the surplus usage of battery.”

Figure 4: I still think that a straight extrapolation of a fitted linear degradation curve would give just as good prediction as the proposed method. If such linear degradation curve is fitted to all measurements, it would also be robust against discontinuously changing capacity.
Response: “Prediction curves before and after turning points” has been revised as “Fitting linear attenuation curve before and after turning point”.

Row 352: Rephrase added sentence part starting with "and battery capacity..." It's meaning is unclear at present.
Response: “and battery capacity does not decrease with the environment” has been revised as “and battery capacity does not decrease with the change of environment”.

Row 356: Explain what is meant with "cycle life budget time"
Response: Explain "cycle life budget time" in line 363.

Row 389-404: Rephrase added section to improve language
Response: “based on correlation vector machine particle filter and autoregressive model fusion is 30%. When SOH is 76, the accuracy of this algorithm is 84%. The accuracy of LiFe PO 4 battery life prediction algorithm based on MIV BP neural network is 43%. The Lithium ion battery life prediction based on correlation vector machine particle filter and autoregressive model fusion. The accuracy of the algorithm is 24%; When SOH is 80,” has been revised as “When SOH is 76%, the accuracy of this algorithm is 84%. The accuracy of LiFe PO 4 battery life prediction algorithm based on MIV BP neural network is 43%. The accuracy of Li-ion battery life prediction algorithm based on correlation vector machine particle filter and autoregressive model fusion is 20%; When SOH is 80%”.

Row 426: the word "leap" is just as unsuitable to use as "jump". I suggest to use the phrase "discontinuous changes" or "non-monotonically decreasing capacity.
Response: The word "leap" has been revised as the phrase "discontinuous changes".

Due to the above mentioned examples of areas that have to be improved, I must recommend that this manuscript must undergo a major revision and I look forward to reviewing an improved version.

Reviewer 3 Report and Authors' Response

The authors have respond to all comments

Response: Thank you for your kind suggestion, I have polished and revised the paper to make it more academic. All the typos have been corrected.

Round  3

Reviewer 1 Report and Authors' Response

The meaning of the acronym SoH still does not appear when the term is first used (now online 62).
The technique presented here is more of a laboratory technique than a truly implantable one, unless it has been demonstrated otherwise.
I would have liked a more international bibliography.
I regret the lack of response to some remarks or the way the plan is presented in the article, but remaining the last reviewer to have pending remarks, I will abide by the majority.

Response: I am sorry that I misunderstood the meaning of the problem. SOH is an abbreviation of “State Of Health”, the sentence “SOH is a byte that must be attached to ensure the normal and flexible transmission of information payload for the network to operate, manage and maintain OAM (Operation, Administration, Maintenance, abbreviation OAM).” on Line 62 has been revised as “SOH (State Of Health) is a byte that must be attached to ensure the normal and flexible transmission of information payload for the network to operate, manage and maintain OAM (Operation, Administration, Maintenance, abbreviation OAM).”
Thank you very much for your review.

Reviewer 2 Report and Authors' response

Thanks again for improving the manuscript and resubmitting it for review. At this point, I merely find minor revisions as per below:

Row 40: Rephrase "at specific current performances", use "at a specific point in time" or similar.
Response: It has been revised as “at a specific point in time”.

Row 42: Wrong use of the word "recycling". Here, I presume you mean "cycling" or "usage"
Response: It has been revised as “cycling”.

Row 46: Incomplete sentence, no capital letter at start.
Response: It has been revised as “Charge”.

Row 62: Rephrase, it is unclear what is meant with ..."additional bytes"
Response: “SOH is the additional bytes that is required to ensure the normal and flexible transmission of information net load for the operation, management and maintenance of the network.” has been revised as “SOH (State Of Health) is a byte that must be attached to ensure the normal and flexible transmission of information payload for the network to operate, manage and maintain OAM (Operation, Administration, Maintenance, abbreviation OAM).”

Row 130: What is meant with capacitor here? Is a capacitor used to discharge a battery? That does not seem likely.
Response: A capacitor battery is actually a capacitor, but its capacity is much larger than that of a normal capacitor. Its external performance is the same as that of a battery. Therefore, it is called a “capacitor battery” and is also called a “super capacitor”.

Row 263: Replace "system" with "current profile" (two instances)
Response: It has been revised as “current profile”.

Table 2: Decimal point missing in third column
Response: It has been revised as “5.1%”.

Row 304, 307, 313 and other places: The use of the word "turning" is not appropriate here, do you mean "difference" or perhaps "change"?
Response: It has been revised as “change”.

Figure 4: Please see my remark from the previous revision. It is still unclear how the proposed methods add significant merits compared to a straight linear extrapolation based on all measured capacity values. See attached figure with an example of such extrapolation.
Response: Figure 4 has been revised.

Row 356: Rephrase "cycle life budget", ill-chosen word. At present, it is unclear what is meant with this.
Response: “the cycle life budget time” has been revised as “budget lifecycle time”.

Row 385: Rephrase or use other word than "reloaded", i.e. "discharged" or "cycled", and omit "filled with liquid".
Response: It has been revised as “cycled”.

Table 6-8: Add unit and/or revise text. If the unit is "number of cycles", then the average accuracy is not 0.17, 0.46 and 0.57%, but an order of magnitude higher (last table, 0.57/ca 0.5 = 9%)
Response: The table is not the average accuracy, but the average error.